# Muscle Delivery of Mitochondria-Targeted Drugs for the Treatment of Sarcopenia: Rationale and Perspectives

**DOI:** 10.3390/pharmaceutics14122588

**Published:** 2022-11-24

**Authors:** Francesco Bellanti, Aurelio Lo Buglio, Gianluigi Vendemiale

**Affiliations:** Department of Medical and Surgical Sciences, University of Foggia, Viale Pinto 1, 71122 Foggia, Italy

**Keywords:** sarcopenia, mitochondrial dysfunction, oxidative stress

## Abstract

An impairment in mitochondrial homeostasis plays a crucial role in the process of aging and contributes to the incidence of age-related diseases, including sarcopenia, which is defined as an age-dependent loss of muscle mass and strength. Mitochondrial dysfunction exerts a negative impact on several cellular activities, including bioenergetics, metabolism, and apoptosis. In sarcopenia, mitochondria homeostasis is disrupted because of reduced oxidative phosphorylation and ATP generation, the enhanced production of reactive species, and impaired antioxidant defense. This review re-establishes the most recent evidence on mitochondrial defects that are thought to be relevant in the pathogenesis of sarcopenia and that may represent promising therapeutic targets for its prevention/treatment. Furthermore, we describe mechanisms of action and translational potential of promising mitochondria-targeted drug delivery systems, including molecules able to boost the metabolism and bioenergetics, counteract apoptosis, antioxidants to scavenge reactive species and decrease oxidative stress, and target mitophagy. Even though these mitochondria-delivered strategies demonstrate to be promising in preclinical models, their use needs to be promoted for clinical studies. Therefore, there is a compelling demand to further understand the mechanisms modulating mitochondrial homeostasis, to characterize powerful compounds that target muscle mitochondria to prevent sarcopenia in aged people.

## 1. Introduction

In recent years, life expectancy has enormously increased all over the world. This has been accompanied by growing health problems related to aging, since the extension of the expected lifespan is unavoidably followed by biological modifications that affect the human body. In particular, age-dependent changes related to muscle mass and function are notably evident as individuals become older and older. The loss of muscle mass during ageing, followed by a decline in physical function and mobility, is defined by the term ”sarcopenia” [1]. According to a recently revised European consensus, sarcopenia is typically characterized by low muscle strength, quantity, and quality [2].

The pathogenesis of sarcopenia is extremely complicated and multifactorial, so the therapeutic approach needs to be multimodal [3]. Even though several age-related diseases (including diabetes mellitus, osteoporosis, neurodegenerative, cardiovascular, and respiratory disorders) facilitate sarcopenia, the homeostasis of muscle mass and strength is maintained through several hormonal and nutritional factors, as well as physical activity levels [4,5,6]. Indeed, sarcopenia develops as the result of a reduced protein intake, associated with failing anabolic pathways modulated by the growth hormone (GH)/insulin-like growth factor (IGF) and vitamin D, and chronic low-grade inflammation [7]. All these findings are related to the impairment of muscle bioenergetics, which mainly rely on mitochondrial metabolism and homeostasis [8]. Most research suggests that modifications in mitochondrial biogenesis, morphology, function, and dynamics may represent the key process in disrupted muscle function and quality [9].

Progression in mitochondrial research and biomedical technology encourages the development of drugs specifically targeted to mitochondria for therapeutic use [10]. Skeletal muscle is an attractive tissue for studying the concept of drug delivery, since it is easily accessible from the bloodstream and includes several specific receptors/transporters that can be used for the selective uptake of molecules [11]. After a description of the importance of mitochondria in the pathogenesis of sarcopenia, this review focuses on mitochondrial drug-delivery systems and muscle-targeted molecules. Considering that mitochondrial medicine is in its developmental stage, we aim to stimulate the progression of muscle mitochondria-directed therapeutics for the management of sarcopenia.

## 2. Age-Related Skeletal Muscle Changes and Sarcopenia

In humans, the reduction in skeletal muscle mass and function begins after the fourth decade [2]. Indeed, the aging process induces a loss of 30% in skeletal muscle mass, with a yearly rate of 0.64–0.70% in women and 0.80–0.98% in men [12,13]. The prevalence of sarcopenia may vary according to different definitions, settings, and sex. Recent meta-analyses reported a prevalence of 11% and 9% in community-dwelling men and women, respectively; the prevalence increased to 51% and 31% in nursing homes, and to 23% and 24% in hospitalized males and females, respectively [14,15].

### 2.1. Age-Related Modifications of Skeletal Muscle Metabolism and Proteostasis

In the human body, skeletal muscle represents the main metabolic tissue since it needs a considerable utilization of oxygen and macronutrients to generate ATP for contraction. In fact, during an intensive activity, skeletal muscle tissue takes account for 60% of the total body oxygen uptake. Furthermore, skeletal muscle regularly disposes of phosphocreatine and glycogen to guarantee partial energy in anaerobiosis [16].

The age-related decrease in skeletal muscle quality is linked to a reprogramming of tissue metabolism, which leads to an altered glucose, lipid, and protein uptake and consumption, with the consequent impairment of ATP production (Figure 1) [17].

The age-dependent disruption of the skeletal muscle metabolism is different between men and women since it is affected by sex hormones [18]. Indeed, changes in body composition characterized by skeletal muscle decline and visceral adipose tissue increase are greater in men, while women exhibit a lower capillarization of type II glycolytic myofibers [19].

Metabolism in old skeletal muscle is affected by the fiber composition, since type I slow-twitch fibers prefer to oxidize fatty acids, while the anaerobic glucose metabolism is favored by type II fast-twitch fibers. The loss of type II fibers is more consistent than type I fiber decline, exerting an impact on tissue metabolism [20]. Furthermore, a reduction in capillarization occurs in aged skeletal muscle, with a consequently reduced nutrient delivery [21,22].

In skeletal muscles enriched with type II fibers, there is a close link between the capillary-to-fiber ratio and muscle fiber size, so these muscles are particularly susceptible to age-related disruption [23]. Enzymes accounting for glycogen metabolism and glycolysis, as well as GLUT4 protein (the muscle-specific transporter-mediating insulin-dependent glucose uptake), are downregulated in aging skeletal muscle [24,25]. Furthermore, several modifications of insulin signaling, which contribute to insulin resistance and an altered glucose metabolism, occur in aged skeletal muscle [26]. Skeletal muscle steatosis in aging mainly relies on changes in lipid uptake and oxidation. Indeed, triglyceride storage in skeletal muscle occurs as a result of an increased uptake and decreased oxidation of palmitate in aged rodents [27,28]. Moreover, changes in proteostasis, characterized by the imbalance between protein synthesis and breakdown, folding and trafficking, in support of excess catabolism that triggers a loss of skeletal muscle quantity and quality, occur in aging [29]. Nevertheless, protein turnover can be influenced by several factors, including nutritional status, physical activity, and insulin sensitivity [30,31,32]. Modifications coming from these factors may somewhat contribute to the ongoing age-dependent impairment of muscle quality [33]. Skeletal muscle from old subjects is no more responsive to the availability of amino acids, losing the capacity to trigger protein synthesis and counterbalance breakdown; this model is referred to as anabolic resistance [34]. The conglomeration of aggregated proteins is a typical feature of proteostasis disruption; this may be triggered by the aging-related rise in oxidative injury [35,36]. Furthermore, the loss of proteostasis is dependent on the alteration of the autophagy–lysosomal and the ubiquitin–proteasomal systems, the two most determinant pathways for protein degradation [37].

### 2.2. Age-Dependent Morphological and Functional Changes in Skeletal Muscle

Most investigations refer to the loss of skeletal muscle mass as the main trigger of sarcopenia (leading to a decrease in muscle strength and performance); nevertheless, modifications in strength and performance may precede the loss of mass in aged people [13].

According to the speed of shortening and myosin ATPase activity, muscle fibers can be classified as fast (type II, white morphology) and slow (type I, red morphology, dependent on high capillarization and the myoglobin content, which provide more oxidative capacity) [38]. The age-related loss of skeletal muscle mass is linked to a reduction in both the number and size of muscle fibers [39]. These changes depend on the age-induced loss of motor units (MUs), which are identified as the muscle fibers innervated by a single soma of an alpha motor neuron located in the ventral horn of the spinal cord. Indeed, the loss of a MU induces fiber denervation and skeletal muscle atrophy. This loss may be compensated with MU remodeling, defined by the reinnervation of denervated muscle fibers by nearby axons; nevertheless, this remodeling is flawed in sarcopenia [40].

Muscle fibers represent approximately 70% of skeletal muscle composition in adult men, but these are decreased to approximately 50% in the elderly, due to an increase in connective tissue (fibrosis) and lipids (steatosis) that accumulate within the muscle (intermuscular adipose tissue, IMAT) and below the fascia [39]. In particular, IMAT is an independent predictor of gait-speed reduction in aging, impairing both muscle strength and muscle metabolism [19,41]. Furthermore, skeletal muscle fibrosis is probably the cause of impaired tissue regeneration [42]. Indeed, fibrosis is the final consequence of several events, including tissue degeneration, recurrent microtrauma, inflammatory cell infiltration, and fibroblast proliferation [29]. Lastly, the size of skeletal muscle fibers tends to reduce with age; although this occurs mostly in type II fibers, a decrease in the diameter of type I fibers has also been reported [43,44].

Skeletal muscle strength and performance are not exclusively dependent on mass and fibers, sustaining the hypothesis that a decrease in skeletal muscle quality—more than quantity—occurs with age [45]. Skeletal muscle quality can be defined as strength (or power) per unit of muscle mass and relies both on the architecture and on the metabolism of skeletal muscle tissue [13,46].

Modifications in the skeletal muscle architecture include changes in fiber number, composition, and size. The main features of skeletal muscle architecture include the cross-sectional area (CSA), pennation angle (PA, measured as the length of the fascicle between the superficial and deep aponeurosis), and fascicle length (FL, measured as the angle between fascicle and deep aponeurosis). While muscle strength is influenced by PA, the muscle shortening velocity is affected by FL [47]. Aging determines a progressive reduction in CSA, PA, and FL [46,48]. This decrease exerts a negative effect on skeletal muscle power, which can be defined as the amount of force generated per unit of time. With respect to the reduction in skeletal muscle function in aging, an average power loss of 72% is more perceptible than the decline in strength in the elderly rather than adults, as the speed of skeletal muscle fiber shortening is reversed on myosin heavy-chain ATPase [49].

## 3. Mitochondrial Involvement in Sarcopenia

Aerobic capacity declines with age, together with changes in skeletal muscle energy metabolism [46]. Effective skeletal muscle bioenergetics rely on mitochondria, and mitochondrial dysfunction is one of the main hallmarks of aging [50]. Mitochondria in skeletal muscle are located below the sarcolemma (subsarcolemmal, SS) or between the myofibrils (intermyofibrillar, IMF); SS and IMF mitochondria are characterized by peculiar morphological and biochemical characteristics [51,52,53]. Age-related changes in skeletal muscle mitochondria involve the morphology, dynamics (fission and fusion), function (bioenergetics and apoptosis regulation), and turnover (biogenesis and mitophagy).

### 3.1. Age-Related Alterations in Morphology and Dynamics of Skeletal Muscle Mitochondria

Research on morphology in aged skeletal muscle describes giant mitochondria with disrupted cristae [54]. Furthermore, compared with the skeletal muscle of young/adults, old SS mitochondria seem fragmented and positioned in a thin layer, while IMF mitochondria appear less reticular [55]. Of interest, a decrease in IMF size was reported in old people, particularly in women more than men, although there were no differences in skeletal muscle size between both sexes [56]. Morphological alterations in old skeletal muscle mitochondria may result from changes in mitochondrial dynamics, characterized by an imbalance that enhances fission rather than fusion [57]. Mitochondrial dynamics can be dysregulated by mtDNA mutations, since old mice expressing a defective mtDNA polymerase gamma exhibited enhanced mitochondrial fission in skeletal muscle [58]. Nevertheless, a higher mitochondrial fusion was described in skeletal muscle from old versus young mice [59]. A change toward mitochondrial fusion rather than fission was further described in the skeletal muscle of old hip-fractured patients [60]. A knock-out of fusion-related proteins (mitofusins, Mfn1/2) in skeletal muscle caused higher mtDNA mutations and tissue atrophy [61]. However, skeletal muscle degeneration and atrophy were described as a result of the deletion of the fission-related protein Drp1 [62]. Taken together, these studies suggest that modifications of mitochondrial dynamics in skeletal muscle and their commitment in sarcopenia need to be clarified.

### 3.2. Mitochondrial Dysfunction and Apoptosis in Old Skeletal Muscle

Dysfunctional mitochondria cause both the exhaustion of ATP and excess of reactive species, with the consequent initiation of damaging cellular pathways. In old skeletal muscle, decreases in mitochondrial activity of the enzymes involved in the tricarboxylic acid cycle, oxygen consumption, and ATP synthesis are reported [63]. Moreover, mitochondrial dysfunction triggers apoptosis, with a negative impact on skeletal muscle quality [64].

Among the worsened mitochondrial functions in old skeletal muscle, the activity of metabolic enzymes (such as citrate synthase) and oxidative phosphorylation (OXPHOS) complexes, protein synthesis, and ATP production rate (mostly caused by an increase in mitochondrial uncoupling) were described [65,66,67,68,69]. However, it is worth noting that mitochondrial function in old skeletal muscle can be preserved with durable and intense physical activity [70,71,72]. To comply with this statement, exercise-mimicking compounds, such as AMP-activated protein kinase or peroxisome proliferator-activated receptor-δ (PPAR-δ) agonists, might act synergistically with mitochondria-targeted therapies to improve muscle quality [73].

Age-dependent reduction in mitochondrial gene expression is described when the transcriptome of old skeletal muscle is compared to young people, although proteomic investigations are controversial, suggesting the need for further studies [74]. Notably, genes related to mitochondrial structure and function are downregulated in old women compared to men, suggesting that females may be more prone to age-dependent mitochondrial impairment in skeletal muscle [75].

In sarcopenia, mtDNA and mitochondrial electron transport chain (ETC) changes are triggered by oxidative stress [76]. Indeed, the highest prevalence of mtDNA deletions is reported in those skeletal muscle fibers exposed to oxidative injury [77,78]. Increased mtDNA deletions are related to modifications of mitochondrial enzymes in old primates and humans [79,80]. An inactive lifestyle in old age is related to mitochondrial dysfunction and oxidative injury in human skeletal muscle, so physical activity may prevent mitochondrial-dependent sarcopenia [80,81]. Induced mtDNA mutations in the skeletal muscle of mice caused a disruption in ETC assembly and function, impairing mitochondrial bioenergetics and ATP homeostasis, and triggering apoptosis and sarcopenia [82]. Dysfunctional mitochondria were also reported in spinal motor neurons from old humans, contributing to the denervation and collapse of skeletal muscle quality [83]. Notably, the denervation of skeletal muscle fibers triggers mitochondrial reactive species even in nearby innervated fibers, indicating a collateral mechanism in sarcopenia [84].

Dysfunctional mitochondria may trigger apoptosis in old skeletal muscle. Indeed, mitochondria from aged skeletal muscle exhibit a high production rate of reactive species and low calcium internalization, with the consequent opening of the mitochondrial permeability transition pore (mPTP), the release of cytochrome c, and DNA fragmentation, all markers of apoptosis [69,85]. Training exercises may reduce the mitochondrial release of proapoptotic proteins and the resultant DNA fragmentation [86,87]. Mitochondrial dysfunction may also induce a caspase-independent apoptotic pathway that contributes to the disruption of muscle quality in aging [88]. The calcium retention capacity was shown to be reduced in skeletal muscle mitochondria from old men, indicating mPTP sensitization to apoptosis [85]. Thus, mitochondria-dependent apoptosis in skeletal muscle represents a potential therapeutic target to counterbalance sarcopenia, as suggested by both in vitro and ex vivo studies [89,90].

### 3.3. Age-Dependent Alterations in Skeletal Muscle Mitochondria Biogenesis and Mitophagy

Alterations in skeletal muscle quality are also dependent on changes in mitochondrial biogenesis. Mitochondrial homeostasis in skeletal muscle is under the control of the peroxisome proliferator-activated receptor-gamma coactivator (PGC)-1α, the master regulator of mitochondrial biogenesis, which is promoted by contractile activity and induces the switching from glycolytic toward oxidative fibers [91]. Nevertheless, an age-dependent decrease in mitochondrial biogenesis may be sustained by the defective response of PGC-1α to exercise [92]. The decreased mitochondrial content in old skeletal muscle may also be dependent on lower PGC-1α expression, which has been described both in slow- and in fast-twitch fibers [54,57,66]. Nevertheless, other studies have described opposite results related to the expression level of the mitochondrial transcription factor A (Tfam), a downstream main PGC-1α transcription factor, in old skeletal muscle [93,94,95].

The limited capacity of senescent skeletal muscle cells to remove injured mitochondria (mitophagy) could be a further cause of mitochondrial alteration. Nevertheless, studies on skeletal muscle from rodents show debated results on mitophagy modulators [96,97,98]. PGC-1α overexpression in skeletal muscle inhibits mitophagy, which appears enhanced in aging [96]. Genes related to mitophagy were described as downregulated in a cross-sectional study on physically inactive frail old women [99]. On the contrary, mitophagy and its regulatory proteins were increased in rodent models of sarcopenia [100,101]. Another study indicated that lysosomal dysfunction may cause an accumulation of disrupted mitochondria in the skeletal muscle of old mice [102]. The muscle-specific deletion of the Mtf2 gene in mice alters autophagy and triggers an adaptive mitochondrial quality control pathway [103]. Controversial results on mitophagy in sarcopenia suggest the need for further research, since this could be a potential therapeutic target. Indeed, the overexpression of the mitophagy regulator Parkin in skeletal muscle attenuates sarcopenia by enhancing mitochondrial content and function [104].

## 4. Muscle Mitochondria-Targeted Therapy for the Management of Sarcopenia

### 4.1. Mitochondria-Targeted Delivery Systems

Mitochondrial delivery strategies can be classified either as referring to the molecular size and type or considering the molecular mechanism [105,106]. According to the first, the best strategy to target mitochondria for the treatment of sarcopenia consists of the use of 1–1000 nm sized particles, which can directly trigger myotubes or inflammatory cells [107,108,109]. According to the latter, passive and active mechanisms are described. Passive targeting relies on the physical and chemical properties of carrier systems, while active targeting refers to specific interactions (ligand–receptor or antigen–antibody) at mitochondrial sites (Figure 2) [110].

#### 4.1.1. Passive Delivery

Several small-sized compounds can be highly localized within mitochondria because of their biochemical and biophysical features (lipophilicity and/or positive charge). Classified as delocalized lipophilic cations (DLCs), these compounds easily cross mitochondrial membranes and locate in the matrix. DLCs include tetraphenylphosphonium (TPP^+^) or its methylated form (TPMP^+^), dequalinium (DQA), and guanidine [110]. DLCs are conjugated to deliver antioxidant compounds, to selectively transport DNA or anticancer agents, sorbitol, metals, and copolymers [110]. Even though DLCs allow for the mitochondrial administration of a specific drug dose, preventing toxicity and resistance, their delivery is limited to electrically neutral and very small conjugates, together with an increased risk of depolarization [10]. Szeto–Schiller (SS) peptides are cell-permeable short peptides (less than 10 amino acids) with antioxidant properties, whose cellular uptake is only dependent on concentration, but not on an electric charge, preventing the risk of depolarization [111]. Liposomes are spherical compounds consisting of phosphatidylglycerol, phosphatidylcholine, and cholesterol, with a hydrophilic core surrounded by a lipid bilayer [112]. Liposomes are nontoxic and can deliver large-sized drugs, including antioxidants, mitochondria-targeted molecules, or even mtDNA [113,114].

#### 4.1.2. Active Delivery

A different strategy to deliver compounds within mitochondria consists of the use of peptides, which are specifically recognized by signal sequences and cleaved off after effective import.

Cell-penetrating peptides (CPPs), such as R8 (RRRRRRRR) and TAT (RKKRRQRRR), are used to enhance the delivery of oligonucleotides, peptides, proteins, and liposomes [115,116].

Mitochondria signal peptides (MSPs) or mitochondria-targeting sequences (MTSs) are normally used to import proteins synthetized in ribosomes within mitochondria [117]. These MSPs or MTSs can be conjugated to nonmitochondrial compounds to form chimeric molecules that are specifically recognized by mitochondrial import machinery, selectively delivering to the intermembrane space, inner membrane, or matrix [117].

Mitochondria-penetrating peptides (MPPs) are artificial compounds based on a CPP strategy, but enriched with positively charged peptides and extra lipophilic amino acids that can efficiently cross the mitochondrial bilayer and interact with the inner mitochondrial membrane [118]. Indeed, compounds covalently conjugated to MMPs are able to mostly accumulate in mitochondria rather than the cytoplasm or nucleus [119].

### 4.2. Mitochondria-Targeted Therapy in Muscle Tissue

To date, exercise is the sole proven therapy for sarcopenia, since it can limit modifications induced by muscle aging [120,121,122]. Nevertheless, several sarcopenic patients are not able to exercise because of clinical complications and/or protracted immobilization. Consequently, the development of compounds that limit the loss of skeletal muscle mass and function is strongly encouraged. To be significantly effective, these compounds should be conveyed using a suitable drug delivery system. A main determinant strategy for developing such compounds is muscle-targeting delivery systems. Among examples of muscle-targeting peptides, the heptapeptide sequence ASSLNIA improves the specificity for binding to skeletal muscle by screening a random phage display library [123]. The 5-polyamidoamine dendrimer (G5-PAMAM) modified with ASSLNIA may synergistically improve skeletal muscle gene delivery [124]. The 12-mer peptide M12, which increases the binding affinity to myoblasts, conjugated with phosphorodiamidate morpholino oligomers may improve muscle function [125].

A promising approach to boost mitochondrial function in muscles consists of increasing intracellular NAD^+^ by inhibiting enzymes that deplete its intracellular levels. The prolonged utilization of MRL-45696, a dual inhibitor of poly(ADP-ribose) polymerases 1 and 2 (PARP1 and PARP2, which consume NAD^+^), improves mitochondrial function in mouse skeletal muscle [126]. The nicotinic acid derivative acipimox, an NAD^+^ precursor, is able to directly enhance skeletal muscle mitochondrial function in humans [127].

Evidence on the efficacy of mitochondria-targeted drug delivery in skeletal muscle was provided in several preclinical studies. Mitoquinone Q, a mitochondria-targeted antioxidant, was able to improve muscle strength and mass in a murine model of cancer cachexia, stimulating beta-oxidation and promoting a shift from glycolytic to oxidative metabolism in muscle fibers [128]. Mito-TEMPOL, a mitochondria-targeted superoxide dismutase mimetic, prevents muscle weakness and wasting via the improvement in mitochondrial function in models of sepsis and uremia [129,130]. The mitochondria-targeted Szeto–Schiller peptide SS-31 was shown to improve exercise tolerance by increasing mitochondrial quality without mitochondrial content in aged mice [131].

## 5. Conclusions and Perspectives

Mitochondria are the most important cellular generators of energy, but also the main source of reactive species. Furthermore, mitochondria regulate cell death. The significance of mitochondrial alterations in the pathogenesis of sarcopenia is sustained through solid investigations, so that these organelles remain attractive therapeutic targets.

Innovative pharmacology is able to produce molecules that can modulate mitochondria in several ways. Compounds can actively or passively enter mitochondria and act as scavengers or substitute molecules. However, several of these molecules need to be tested in vivo for the treatment of sarcopenia. Preclinical experiments strongly advise for their potential efficacy in preserving mitochondrial quality and function, counterbalancing oxidative stress and preventing mitochondrial apoptosis. The development of molecules targeted to skeletal muscle mitochondria could overwhelm several challenges associated with actual therapies, increasing the efficacy and decreasing toxicity. Even though mitochondrial medicine is developing, current applications in the treatment of sarcopenia support future clinical studies.

## Figures and Tables

**Figure 1 pharmaceutics-14-02588-f001:**
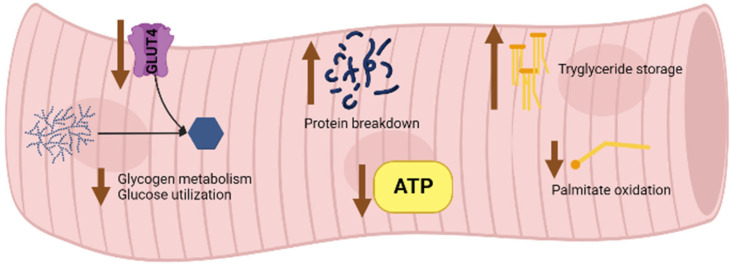
Age-associated altered metabolism in skeletal muscle.

**Figure 2 pharmaceutics-14-02588-f002:**
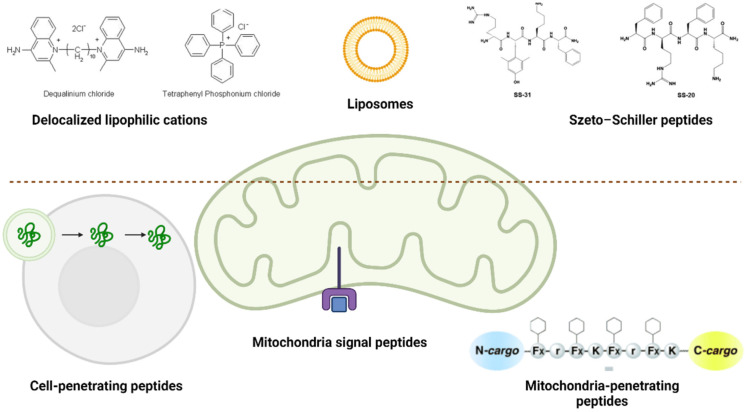
Passive (up the dotted line) and active (down the dotted line) mitochondria delivery strategies.

## Data Availability

Not applicable.

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
