# Peer review of "Muscle Delivery of Mitochondria-Targeted Drugs for the Treatment of Sarcopenia: Rationale and Perspectives"

_pharmaceutics, 2022, doi:10.3390/pharmaceutics14122588_

Round 1

Reviewer 1 Report

Regarding the manuscript (pharmaceutics-2036143) entitled:

Muscle delivery of mitochondria-targeted drugs for the treatment of sarcopenia: rationale and perspectives

General comment

The study presented the most recent evidence on mitochondrial defects that are thought to be relevant in the pathogenesis of sarcopenia and that may represent promising therapeutic targets for its prevention/treatment. The manuscript, in general, is well written with sufficient data that proved the aim of the study.

Comments:

-          I recommend presenting the relation between particle size and treatment of sarcopenia.

Author Response

We thank the reviewer for his comment. Accordingly, in the revised manuscript we referred to studies which used particles between 1 and 1000 nm (nanoparticles) to treat sarcopenia (page 6, lines 256-259).

Reviewer 2 Report

In this clearly written review, Bellanti and colleagues offer an interesting and important overview of changes in skeletal muscle function and mass observed during the ageing process. Mitochondrial involvement in this sarcopenia is highlighted by the reviewed literature, and the authors thus identify mitochondria as attractive therapeutic targets. Accordingly, they offer a brief overview of reported strategies to deliver drugs specifically to the mitochondria of skeletal muscle. Although mitochondrial changes are almost certainly associated with age-related muscle dysfunction, it remains possible that these changes are adaptations to declining (basal) energy expenditure by skeletal muscle fibres. If this were the case, then boosting mitochondrial activity pharmacologically will likely be in vain, as there would be no energy demand for such increased activity. Consistent with this possibility, the authors state explicitly that ‘exercise is the sole proven therapy for sarcopenia’ and cite positive effects of physical activity on mitochondrial quality (for example, lines 190-191, 204-205, 216). ‘Mitochondria-improving’ drugs may thus only be effective when administered in combination with ‘exercise mimetics’ that stimulate myocellular energy demand. The authors may wish to include this argument somewhere in their paper.

Author Response

wWe thank the reviewer for his valuable comment. We included the suggested argument in the new version of the manuscript (page 5, lines 192-195).